# When Incorporated into Fruit Sorbet Matrix, Are the Fructans in Natural Raw Materials More Beneficial for Bone Health than Commercial Formulation Added Alone?

**DOI:** 10.3390/ani12091134

**Published:** 2022-04-28

**Authors:** Kinga Topolska, Marek Bienko, Paweł Ptaszek, Adam Florkiewicz, Radoslaw P. Radzki, Agnieszka Filipiak-Florkiewicz

**Affiliations:** 1Department of Plant Product Technology and Nutrition Hygiene, Faculty of Food Technology, University of Agriculture in Krakow, al. Mickiewicza 21, 31-120 Krakow, Poland; agnieszka.filipiak-florkiewicz@urk.edu.pl; 2Department of Animal Physiology, Faculty of Veterinary Medicine, University of Life Sciences in Lublin, Akademicka 12, 20-033 Lublin, Poland; marek.bienko@up.lublin.pl (M.B.); radoslaw.radzki@up.lublin.pl (R.P.R.); 3Department of Fermentation Technology and Microbiology, Faculty of Food Technology, University of Agriculture in Krakow, al. Mickiewicza 21, 31-120 Krakow, Poland; pawel.ptaszek@urk.edu.pl; 4Department of Food Analysis and Quality Assessment, Faculty of Food Technology, University of Agriculture in Krakow, al. Mickiewicza 21, 31-120 Krakow, Poland; adam.florkiewicz@urk.edu.pl

**Keywords:** calcium hypoalimentation, femur, fructans, rats, bone markers

## Abstract

**Simple Summary:**

Because of the serious consequences of osteoporosis, the prevention of this disease, which is also known as “the Silent Epidemic”, is considered essential to the maintenance of human health. It is crucial to maximize bone mass and strength during childhood and adolescence to reduce the risk of osteoporosis in later periods of life. The optimization of nutrition is one of the most important factors. Special attention should be paid to ensuring sufficient calcium intake, which is a problem in several population groups, and its bioavailability. We assessed the effect of fructans, taking into consideration their source (Jerusalem artichoke, yacon, commercial preparation) and the form of their addition to the diet (alone or as an ingredient of strawberry sorbet). A growing-female-rat model was used to assess the potential functional properties of these non-digestible carbohydrates in/without the “fruit matrix”, during calcium hypoalimentation. We showed that fructans have a protective effect during the inadequate supply of Ca, but their action is related to their source as well as the dietary matrix. It is important to look for new preventive strategies, and our paper gives an insight into various fructan sources, depicting their action and how these natural dietary components can be exploited.

**Abstract:**

We assessed the extent to which fructans from various sources and added in various forms (raw materials in diet alone or incorporated into a strawberry matrix) differ in their effectiveness towards selected parameters related to bone health under calcium hypoalimentation in growing female Wistar rats. The aim of this study was to evaluate the levels of selected parameters involved in calcium metabolism, in response to a 12-week restriction of Ca intake: serum ions (Ca, Mg, P); the activity of alkaline phosphatase—using a BS 120 analyzer; the markers of bone turnover (osteocalcin, CTX; using a Rat-MidTMOsteocalcinEIA Kit and RatLapsTMEIA, respectively); and the bone mineral content (BMC) and density (BMD), using a Norland Excell Plus Densitometer. Among the examined markers, the CTX concentration increased dramatically under calcium hypoalimentation. The presence of Jerusalem artichoke (independently of the form of addition) and yacon root powder (with strawberry sorbet matrix) in the rats’ diet led to a significantly lower CTX concentration than was observed in the low-calcium control group. The type of fructan influenced the bone mass content. When fructan was added to the low-calcium diet as an ingredient of sorbet, it exerted more pronounced effects on the biochemical parameters of bone metabolism than when added alone, in the growing-female-rat model.

## 1. Introduction

Osteoporosis is one of the most devastating epidemics of the 21st century. This disease affects ca. 200 million people worldwide, with a significant impact on rates of morbidity and mortality. Osteoporosis is named “the Silent Epidemic” because it typically has no symptoms until a bone fracture occurs [1].

Bone is a highly active tissue, which is continually formed/resorbed during the life span. Its morphology and functions are maintained by bone remodeling processes [1,2].

To reduce the risk of osteoporosis in later periods of life, it is crucial to maximize peak bone mass and bone strength during childhood and adolescence [3,4]. Bone development requires sufficient amounts of many nutrients, but calcium has received the greatest attention as its principal component. A low-calcium diet or/and lowered Ca absorption are the most important problems related to bone loss [5]. Unfortunately, a large proportion of the world’s population consumes a low-calcium diet. As an example, in the United States, low calcium intake is recognized as a major public health concern [6,7]. The Polish population also does not consume sufficient Ca to meet current recommendations. Deficiency in this mineral is observed not only in the elderly [8,9], but also among children [10].

Meanwhile, bone mass and bone density are determined by several concurrent factors, including not only genetics, hormones, and physical activity, but also nutrition [11]. The adequate intake of bone nutrients is a significant factor in the expression of a given potential and for the maintenance of bone health at all stages of life [11,12].

Bone biomarkers (i.e., osteocalcin, CTX, alkaline phosphatase), including formation, resorption, and regulation, are released during bone remodeling processes [13]. Combined with the measurement of bone mineral density (which is the “gold standard”), the clinical applications of bone biomarkers have provided comprehensive information for the diagnosis of osteoporosis [13].

Because of the increase in healthcare costs and average life expectancy, the many people seek ways to improve their physical condition and, in turn, develop a higher quality of life. This is why functional food science has gained momentum, in response to the health problems of the human population [14]. The concept of “functional food”, in the form of the components of our diet rather than pills, is regarded as the possible solution to chronic non-infectious diseases such as obesity, diabetes, some types of cancer, and osteoporosis [14].

The promotion of the health potential of such foods is based on the presence of essential bioactive compounds (i.e., minerals, vitamins, dietary fiber) [15]. Among these, inulin-type fructans have shown great promise as a dietary intervention strategy. Fructans are polymers of fructose featuring β-(2-1) and β-(2-6) linkages with a glucose molecule at the terminal position. Previous studies have shown that these compounds are beneficial to human health and offer an approach to the treatment of some diseases [16,17].

These compounds could promote bone health through their positive effects on mineral retention, leading to the achievement of increased peak bone mass and bone mass conservation during ageing [18,19,20]. The data related to inulin’s significant roles suggest that it is capable of improving the absorption of Ca through passive diffusion, bolstering the absorption of this mineral (via ion exchange), and expanding the colon’s absorption surface due to the stimulation of cell growth. Moreover, fructan increases Ca solubility, stimulates levels of calcium-binding protein expression, and increases useful microorganisms. Furthermore, calbindin levels are enhanced and transcellular active calcium transport is stimulated by fructan. An intake of 8–10 g inulin per day supports total body bone mineral content and density in adolescents (through its known mechanisms of action) [21].

Thus, the investigation of fructan’s osteoprotective effect should be one of the main directions of scientific research work concerned with the impact of these carbohydrates on bone health.

Many practical observations of the benefits of fructans for health have been obtained using purified molecules or plant extracts, whereas many natural raw materials of plant origin and their significance as “a whole” (including the “matrix”) are often overlooked [20,22]. This is a paradox because these compounds are not present in our daily diet alone, but as components of cereals, fruits, and vegetables. The dietary “matrix”, dose, and conditions belong to the factors affecting fructan’s action [22]. Moreover, there is evidence that long-chain inulin and oligofructose enhance femoral Ca content, bone mineral density, and its retention through enhanced calcium absorption and suppressed bone turnover rates, but they are not bone-promoting under all conditions [23].

Our purpose was to examine the effect of different natural fructan sources added to the diet alone or as components of strawberry sorbet on the parameters related to bone health, as compared to commercial formulations of these compounds.

We hypothesized that the fructan’s effect on the bone parameters would be more beneficial when it was a component of a raw material and incorporated into the diet as a sorbet ingredient.

## 2. Materials and Methods

### 2.1. Animal Study

The study was carried out according to a protocol approved by the Local Ethics Commission in Krakow (no 119/2011). All institutional and national guidelines for the care and use of laboratory animals were followed. Sixty-four female, 5-week old Wistar rats with a mean body weight of 126 g were housed in a controlled environment (20–23 °C, 12 h/12 h day:night) in common cages (two animals per cage). After seven days of acclimatization, animals were randomly divided into experimental groups (described briefly in Table 1), as follows:Validation model, with two groups of rats fed a diet with recommended Ca dose (RC) [24] or Ca-deficient diet (LC) for 12 weeks;Treatment, with six groups of rats fed a low-calcium diet containing 8% fructans, for 12 weeks: LC-JA, LC-Y, and LC-F (raw materials added to the diets alone), as well as LC-JAS, LC-YS, and LC-FS (raw materials added to the diets as a components of strawberry sorbets).


Characteristics of the research material are described in detail in our previous papers [19,25]. The pulp from Jerusalem artichoke tubers and all the sorbets were added to the rats’ diets in freeze-dried form. The diet formulations were calculated to provide equal contents of calcium and fiber, including the amount of fructan (Appendix A).

The rats were allowed free access to the food and deionized water throughout the experiment period. Water and diets were replenished daily. The weight gain and the consumption of the diet were monitored once a week, and no significant differences (*p* > 0.05) were noted. At the end of the experiment, rats were euthanized according to the ethical guidelines for animal experimentation.

### 2.2. Sample Collection and Storage

Blood samples were collected from each animal (5 ± 1.0 mL) and stored at −80 °C until biochemical analysis.

Femurs were separated from adjacent tissue, cleaned, wrapped in saline-soaked gauze, and stored at −80 °C until testing.

### 2.3. Biochemical Parameter Assays

Blood samples were analyzed with standard laboratory methods. Serum ionized calcium (calculated based on total calcium level determination), magnesium and phosphorus concentration, as well as alkaline phosphatase activity, were quantified (in duplicate) using a BS 120 auto chemistry analyzer (Shenzhen MindrayBio-Medical Electronics Co., Ltd., Stamar, Poland, authorized distributor).

The serum levels of osteocalcin (OC) were estimated using a Rat-MidTMOsteocalcinEIA Kit (Immunodiagnostic Systems), and C-telopeptide degradation products from type I collagen (CTX) were determined (in duplicate) by RatLapsTM EIA (Immunodiagnostic Systems) (an enzyme-linked immunosorbent assay).

### 2.4. Bone Densitometric Measurements

The bone mineral density (BMD) and bone mineral content (BMC) of the isolated femora were established by a Norland Excell Plus Densitometer (Fort Atkinson, WI, USA) equipped with Illuminatus DXA Software v.4.5 with a Small Animal Scan option. The measurements were performed using the following parameters: scout scan speed 100 mm/s, resolution 3.0 × 3.0 mm; measurement scan speed 10 mm/s, resolution 1.0 × 1.0 mm. The region of interest (ROI) after the scout scan was defined manually. The densitometer was calibrated using quality assurance phantoms (QA-Phantom) provided by the manufacturer and performed in agreement with set procedures before every measurement series.

### 2.5. Statistical Analysis

Data were presented as mean ± SD. Differences between RC and LC groups were analyzed by the Student’s *t*-test for unpaired samples. Results from treatment groups were subjected to a two-way analysis of variance, where factor 1 was the type of fructan source, and factor 2 was the form of fructan source (alone or in the sorbet). A post hoc Duncan test was used. Differences were considered significant at *p* < 0.05. All calculations were performed with statistical software package Statistica 9.1 (StatSoft Inc., Tulsa, OK, USA).

For a better illustration of differences between experimental groups, a principal component analysis (PCA) was also performed in two versions. The first version was performed taking into consideration the way the Wistar rats were fed (i.e., the form of addition of every fructan source) in two ways: (1) Groups fed low-calcium diets (without RC group); (2) all experimental groups. The second approach was performed using all examined parameters, in three ways, including validation model (RC and LC groups), groups fed low-calcium diets (without RC group), and all experimental groups. The PCA method included a cut-off analysis based on the determined eigenvalues. On this basis, two principal components with the highest values were distinguished. For these components, biplot diagrams were prepared, which were used for further analysis and interpretation of the results. A correlation analysis was also performed and the significance of the obtained correlation coefficients was analyzed using the Spearman test [26]. The PCA analysis was performed with *gnu R* software [27].

## 3. Results

### 3.1. The Changes in Dependence on Dietary Calcium Dose of Examined Parameters

Table 2 and Table 3 report the changes in the examined parameters in the young female rat groups as a function of the composition of their diets, i.e., different dietary Ca doses (validation model, Table 2), different fructan sources, and the form in which the fructan was added to the diet (experimental model, Table 3). In turn, Table 4 gives the results of the two-factorial statistical analysis and indicates which dietary treatment or combination of treatments significantly affected the examined parameters.

We did not observe significant changes in the levels of ionized calcium and magnesium in the blood serum of the animals fed a calcium-restricted diet compared with the control group (Table 2). Simultaneously, the phosphorus concentration was significantly (*p* < 0.05) elevated in the LC group.

Among the examined markers of bone turnover, calcium hypoalimentation did not impair ALP or OC levels. 

However, it was noted that the CTX concentration increased dramatically (by 62%). When comparing the RC and LC groups, the BMD and BMC were not statistically differentiated (Table 2). 

### 3.2. The Effect of Fructan-Enriched Diets 

In Table 3, we show the results obtained for the experimental groups in comparison to the Ca-deficient control group (LC), except for the OC and CTX. There parameters are presented on Figure 1.

Table 4 summarizes the results of the statistical analysis (two-factorial analysis of variance). The concentrations of calcium, magnesium, and phosphorus in the blood serum were dependent on the fructan type (source). Additionally, in the case of magnesium, the form of fructan addition (with a raw material or as a component of a strawberry sorbet matrix) was also important (Table 4).

The female rats fed a diet containing Jerusalem artichoke, both in raw material form and in the sorbet were characterized by a markedly higher Ca^2+^ serum level compared to the LC, LC-Y, LC-YS, and LC-FS (Table 3). Moreover, the level of magnesium was significantly enhanced in the groups fed the diets enriched in fructans in raw material form (LC-JA; LC-Y; LC-F) or the diet with yacon-containing sorbet (LC-YS), in comparison to the results obtained for the LC group. By contrast, the P concentration was markedly lower in all the experimental groups (except for LC-F) than in the low-calcium group (Table 3).

We also observed that the form of the fructans, as well as interactions between two experimental factors, affected the alkaline phosphatase activity (Table 4). This parameter was statistically increased in the animals fed a diet containing Jerusalem artichoke or yacon root powder (compared to the other groups). In parallel, the lowest level of this marker (*p* < 0.05) was noted in the LC-YS animals compared to the LC and all the groups fed with the diets enriched in raw materials (Table 3).

Feeding the female rats with fructan-containing diets led to a decrease in OC serum levels (Figure 1).

The animals from the LC-YS and LC-FS groups were characterized by significantly lower values of this parameter than the LC group (Figure 1). The presence of Jerusalem artichoke in the rats’ diet (independently of the form of addition) and the rats’ consumption of yacon root powder with the strawberry sorbet matrix led to a significantly lower CTX concentration than was observed in the control LC group. The two-factorial analysis of variance showed that the fructan type markedly influenced the bone mass content (Table 3). A significantly higher BMC was shown in the animals fed with the LC-Y and LC-JAS diets compared to the LC-F group. No significant changes in BMD were noted among the treatment groups.

### 3.3. PCA Analysis

The evaluation of dietary intervention effectiveness is combined with the analysis of changes in the levels of several parameters. These parameters show a high variability and, simultaneously, a strong correlation. For a better understanding and interpretation of the obtained results, appropriate statistical tools should be used. Therefore, principal component analysis (PCA) was applied. Two versions of PCA were used. The first was focused on dietary intervention (the presence of fructan), while the second was concerned with the examined parameters. The PCA results for the first version of the analysis are presented in Figure 2a,b. The second version is presented in Figure 3a–c and Figure 4. According to the results from the first version of the PCA—options 1 and 2—two reduced variables were obtained, explaining as much as 95.07% and 95.19%, respectively (Figure 2a,b).

It was observed that the differences between the RC and LC groups were explicit. In turn, analyzing the effect of the fructan source, significant differences were noted for the LC-Y in relation to the LC group (Figure 2a,b). We did not find relevant differences between the LC-JAS, LC-YS, LC-FS, LC-F, and LC-JA groups. With regard to the second version, with limitations on the data numbers, only the two first variables were considered in the subsequent analysis, explaining 55.42%, 47.63%, and 47.26% of the variability, respectively (Figure 3a–c). In option 1 of the PCA analysis (Figure 3a), it was shown that the parameters BMC and serum magnesium were better described by the second reduced variable (positive correlation).

A similar relationship was observed in the case of the BMD. In turn, the CTX and OC were described by the first reduced variable. The ALP, in turn, was described by the combination of both the first and second reduced variables (negative correlation). A negative correlation was also shown for the ionized calcium parameter. It should be noted that the BMC, BMD, CTX, OC, and ALP parameters were located near to the edges of the unit circle (the circle of radius 1), and, therefore, they were all well represented by these reduced variables, forming the coordinate system.

Figure 3b presents the results of the second PCA analysis version—option 2. The four vectors are characterized by the largest length, i.e., BMD, BMC, OC, and P. Their ends are located nearest to the circle with a radius of 1, and, therefore, they are all described by the reduced variables forming the coordinate system.

The angle between the BMD and BMC vectors is very small, which means that these parameters were well correlated. The projections of the vectors on the individual principal components mean that there was a negative correlation with the first reduced variable and a positive correlation with the second variable. It should be noted that the projection on the first reduced variable had a higher absolute value than the projection on the second variable. This arrangement of vectors indicates that they formed a homogeneous group, which was mainly represented by the first reduced variable. The opposite was observed for the CTX vector, which was slightly correlated with the other primary variables. The correlation between this vector and the first reduced variable was positive and not very strong (low value of the first coordinate of the vector end), whereas and, in the case of the second reduced variable, the correlation was also positive and strong (high value of the second coordinate of the vector end). It can be concluded that as the only primary variable, CTX was well represented by the second component. The serum magnesium parameter was well described by the first reduced variable and no significant correlation was found for the other parameters. The ALP was well described by the second reduced variable (in this case, it was also the lack of strong correlation with other parameters). The ionized calcium was, in turn, satisfactorily described by both reduced variables.

In the Figure 3c (PCA version 2, option 3), the first two reduced variables, representing 27.25% and 20.01% of the variance, are presented. The four vectors (BMD, BMC, CTX, and P) were characterized by the longest length, and their ends were located nearest to the circle with a radius of 1. Thus, they were all efficiently described by these variables, forming the coordinate system.

The angle between the BMD and BMC vectors was very small; this means that these parameters were well correlated. The projections of the vectors on the individual principal components showed a negative correlation (with regard to the first reduced variable) and a positive correlation (concerning the second reduced variable). It should be noted that the first reduced variable projection had a higher absolute value than the second. This arrangement of vectors indicates that they formed a homogeneous group, which was mainly represented by the first reduced variable. In the case of the CTX, a completely different direction was observed. This vector was slightly correlated with the other primary variables, as evidenced by the angle of its inclination toward the other primary variables (close to the right angle: BMD, BMC, and Mg^2+^). The correlation of this vector was positive and not very strong in the case of the first reduced variable (according to the low value of the first coordinate of the vector end), whereas it was positive and strong regarding the second reduced variable (according to the high value of the second coordinate of the vector end). It can be concluded that the CTX, as the only primary variable, was well represented by the second reduced variable. A very small angle was observed between the P and OC vectors, which indicates their good correlation. The projections of these vectors on individual reduced variables testified to the positive correlation between the individual vectors. In addition, the values of the first and second reduced variables were similar, which means that these primary variables were well represented by both the first and the second reduced variable. The magnesium parameter is well described by the first reduced variable. The small angle between the ALP and ionized-calcium vectors means that a strong correlation was found between these parameters. In this case, we observed that the ALP was better described by the second reduced variable than the ionized calcium (described by both reduced variables).

Figure 4 shows the correlations between the examined parameters. The part above the diagonal represents the raw materials (LC-F, LC-Y, and LC-JA) and the part below the diagonal represents the sorbets (LC-FS, LC-YS, and LC-JAS). In the case of the raw materials (LC-F, LC-Y, and LC-JA), significant correlations between the CTX and P (positive) and between the CTX and ALP (negative) were observed. Moreover, the densitometric parameters were strong and positively correlated (BMC, BMD). In the case of the sorbets (LC-FS, LC-YS, and LC-JAS) as the fructan sources, more relations between the parameters were found. The concentration of Ca^2+^ was correlated with Mg^2+^ (positive), whereas the level of phosphorus was significantly positively correlated with the OC as well as the CTX. The OC and CTX were also positively (*p* < 0.05) correlated. A strong positive correlation was also observed in the cases of the BMD and BMC.

## 4. Discussion

The skeleton plays several important roles, including mechanical support for stature and locomotion, the protection of organs, and control over mineral homeostasis. It is crucial to maintain a healthy skeleton across the lifespan [28]. Therefore, two main strategies exist for promoting good bone quality. In childhood and adolescence, the best strategy is to engage in objectives that promote the attainment of peak bone mass (PBM). After the achievement of PBM, the strategy should be focused on reducing bone mass loss [6]. However, many factors contribute to the attainment of maximum peak bone mass and strength during the bone accrual period, which may respond differently to Ca levels [4]. Bone is highly affected not only by nutritional calcium but also by several other factors, including physiological conditions or pharmacological treatments [29]. In our study, growing female rats were fed a low-calcium (60% of recommended value) diet (LC group) to show the consequences of Ca hypoalimentation for biochemical and densitometric bone markers. We observed a lack of statistically significant changes in the values of Ca^2+^ after 12 weeks of dietary treatment between the LC and RC animals. By contrast, Kasukawa et al. [30] reported that a two-week period of experimentation was sufficient to observe that a low-calcium diet decreased serum calcium concentrations in growing mice, compared with animals fed a diet with a normal dose of this mineral. According to Kruger et al. [3], one of the ways to improve calcium absorption and, in turn, maximize peak bone mass, is to enrich the diet with fructans. Indeed, we showed a significant (*p* < 0.05) change in serum calcium, magnesium, and phosphorus, which was dependent on the type of fructan. The rats fed a diet containing Jerusalem artichoke (as a raw material or in the sorbet) were characterized by markedly higher Ca^2+^ serum levels compared to the low-calcium group. Coudray et al. [31] showed only a tendency to increase blood calcium concentrations with inulin intake (+2%), whereas Bryk et al. [32] did not observe statistical differences in the values of this parameter among groups.

Molecular markers are a valuable tool for the evaluation of bone remodeling processes. Among these processes, serum alkaline phosphatase plays an important role in osteoid formation and mineralization [33], being a component of the cell membranes of many tissues (with the highest concentrations in the osteoblasts, liver, and kidneys) [34]. This parameter was significantly higher in the group of animals fed a diet enriched in Jerusalem artichoke as well as yacon (both as raw materials). Meanwhile, the addition of yacon as a component of sorbet caused a decrease in ALP levels, compared to the LC group. The changes in this parameter could indicate the disturbances in calcium homeostasis. According to Kim et al. [35], ALP is elevated in diseases of the skeletal system, and is associated with increased osteoblast activity and bone remodeling. It is important to remember that ALP assays are helpful but are of limited value because they do not specifically reflect bone function and bone turnover. For this reason, in studies concerned with Ca metabolism and bone quality, other markers should be also used. One of these markers is osteocalcin—a noncollagenous calcium-binding protein hormone, exclusively synthesized by osteoblasts, odontoblasts, and hypertrophic chondrocytes [33,36]. High OC concentrations demonstrate a good correlation with increases in bone mineral density during bone formation. Therefore, OC can be used as a preliminary biomarker of bone formation [36]. Despite the lack of differences between the RC and LC groups in terms of their OC values, feeding animals with fructan-enriched diets led to a decrease in these marker levels, with statistical significance (*p* < 0.05) observed in the LC-YS as well as the LC-FS group compared with the LC group. A trend toward higher osteocalcin levels in inulin-fed rats was observed by Jamieson et al. [18]. In fact, serum levels of osteocalcin have been reported to be increased in patients with metabolic bone problems [37]. CTXs are degradation products of Type 1 collagen of bone generated by the activity of the enzyme cathepsin K [33]. Measurements of this parameter are indications of bone resorption [3]. In this study, we noticed that bone modeling was severely altered in the Ca-restricted group (LC vs. RC animals) as shown by the dramatic increase in serum CTX levels. This finding agrees with those of other studies, which reported increased bone resorption in Ca-deficient young rats [4,38,39]. A higher level of CTX was determined in the Ca-restricted group (compared to the animals fed a diet with the recommended Ca dose), which may suggest an enhancement in bone resorption. The intake of diets enriched in fructan sources (LC-JA, LC-JAS, and LC-YS) led to a significant decrease in CTX concentrations compared to the LC group (*p* < 0.05). Similar observations were made by Nzeusseu et al. [40]. The authors showed a significant decrease in serum CTX levels in fructan-fed groups, compared to control animals. An enhancement the intestinal absorption of Ca in rats consuming fructans led to reduced bone turnover. Zafar et al. [41] observed that the bone resorption rate was significantly decreased by nondigestible oligosaccharides. Kruger et al. [3] also reported a significant decrease in urinary CTX-I excretion in growing rats fed a diet with inulin for 4 weeks. By reducing the remodeling space and the number of bone resorptive sites, fructans could allow secondary mineralization to catch up and lead to a shift in BMD profile. Many factors, including diet, may affect bone tissue quality and bring about a decrease or increase in bone mineral density and bone mineral content, which are the main parameters related to bone condition [4]. It was reported that high bone-resorption markers with low femoral BMD are more likely to predict fractures than those with only low BMD [42]. Viguet-Carrin et al. [4] noticed that the BMD (with other bone-quality parameters) responded to calcium intake. As anticipated, low Ca intake (compared to an adequate intake), had a detrimental impact on peak bone mass accrual (−29% for BMD), due to the hormonal disruption implied by the mineral metabolism. No harmful effects of high calcium on bone modeling were observed in this short-term study. Meanwhile, in our study, we did not observe such changes in BMD between the validation groups after 12 weeks of feeding. Kasukawa et al. [30] showed that wild-type mice increased their femoral bone mineral density during 2 weeks of pubertal growth, by 35 and 7%, when fed normal- or low-calcium diets, respectively. This indicates that during Ca^2+^ deficiency, bone accretion is impaired. Measuring ex vivo bone mineral density and bone mineral content, we showed that BMC was significantly higher in the LC-Y and LC-JAS groups compared to the LC-F group. Lobo et al. [43], suggested that the action of fructans is dependent on their degree of polymerization (DP), with better results obtained for inulin (DP > 23). Similar observations were made by Nzeusseu et al. [40], who observed an increase in the BMC of excised femurs only in the inulin group.

Principal component analysis is a well-known approach to producing interpretable overviews of the main information in multivariate datasets. It can generate fewer principal components (reduced variables) that are independent of the original variables but show linear combinations and, simultaneously, explain most of the features of the aboriginal data [44,45]. Our analysis (version 2) showed a strong correlation between the BMC and the Mg^2+^, as well as between the BMD and the Mg^2+^. In turn, the PCA analysis performed for all the experimental groups (option 2), demonstrated a good correlation between the P and the OC. A similar relationship was shown between the ALP and the ionized calcium. Based on the PCA results of our previous study [19], we concluded that the variability in bone quality was affected by the fructan source, and also by the form of its addition to the diet.

## 5. Conclusions

Our study is unique in that it compares the different fructan sources and forms of addition (different matrix) in the same model—growing female rats. The investigation of the osteoprotective effect of fructan was our main aim. We clearly showed that fructans from natural sources, particularly when added to the diet as ingredients of strawberry sorbet, had a more beneficial effect (evidenced by the significant decrease in CTX levels) under calcium hypoalimentation, compared to the commercial formulation. Among plants rich in fructans, special attention should be paid to Jerusalem artichoke. We evidenced that this raw material exerted a positive effect on the bone parameters when added to the diet alone, as well as in the sorbet matrix.

## Figures and Tables

**Figure 1 animals-12-01134-f001:**
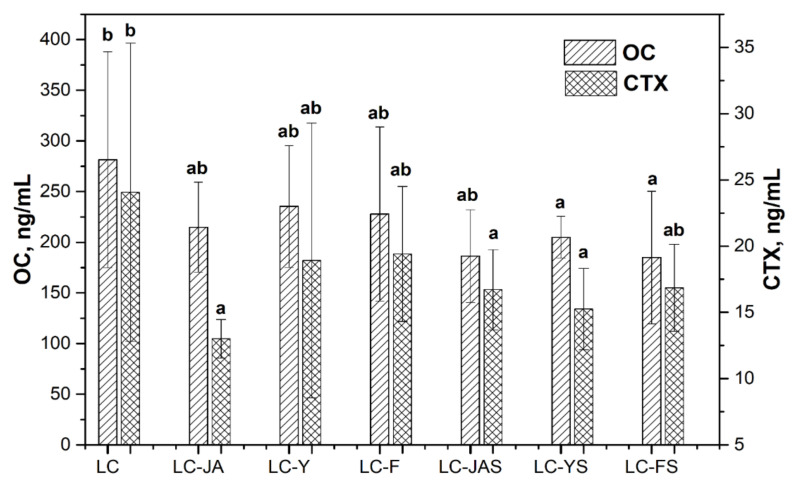
Changes in OC and CTX levels in rat serum after 12 weeks of feeding animals with the low-calcium diets enriched with fructans compared to LC group. a,b—different small letters mean significant differences among low-calcium groups, separately, for each parameter (*p* < 0.05).

**Figure 2 animals-12-01134-f002:**
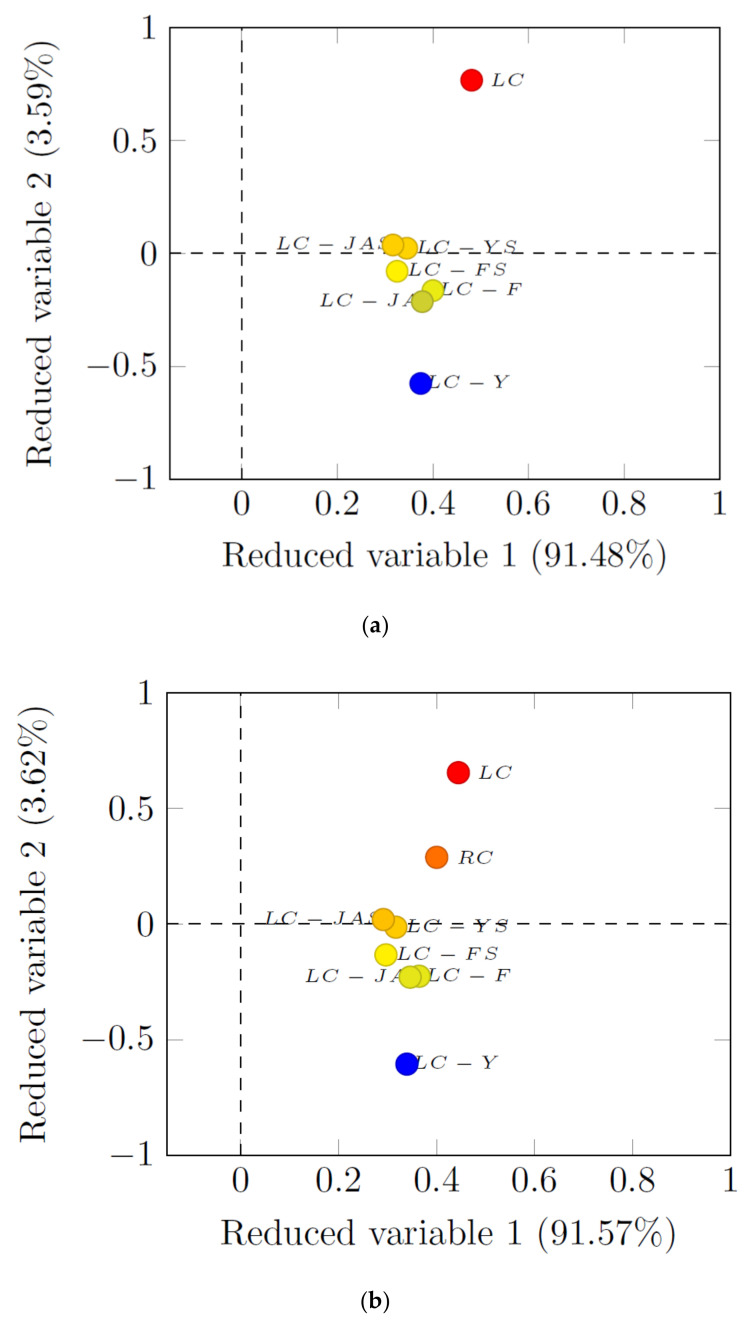
(**a**) Principal component analysis (first version, option 1) with distribution of analyzed samples (LC-groups). (**b**) Principal component analysis (first version, option 2) with distribution of analyzed samples (all groups, including RCD).

**Figure 3 animals-12-01134-f003:**
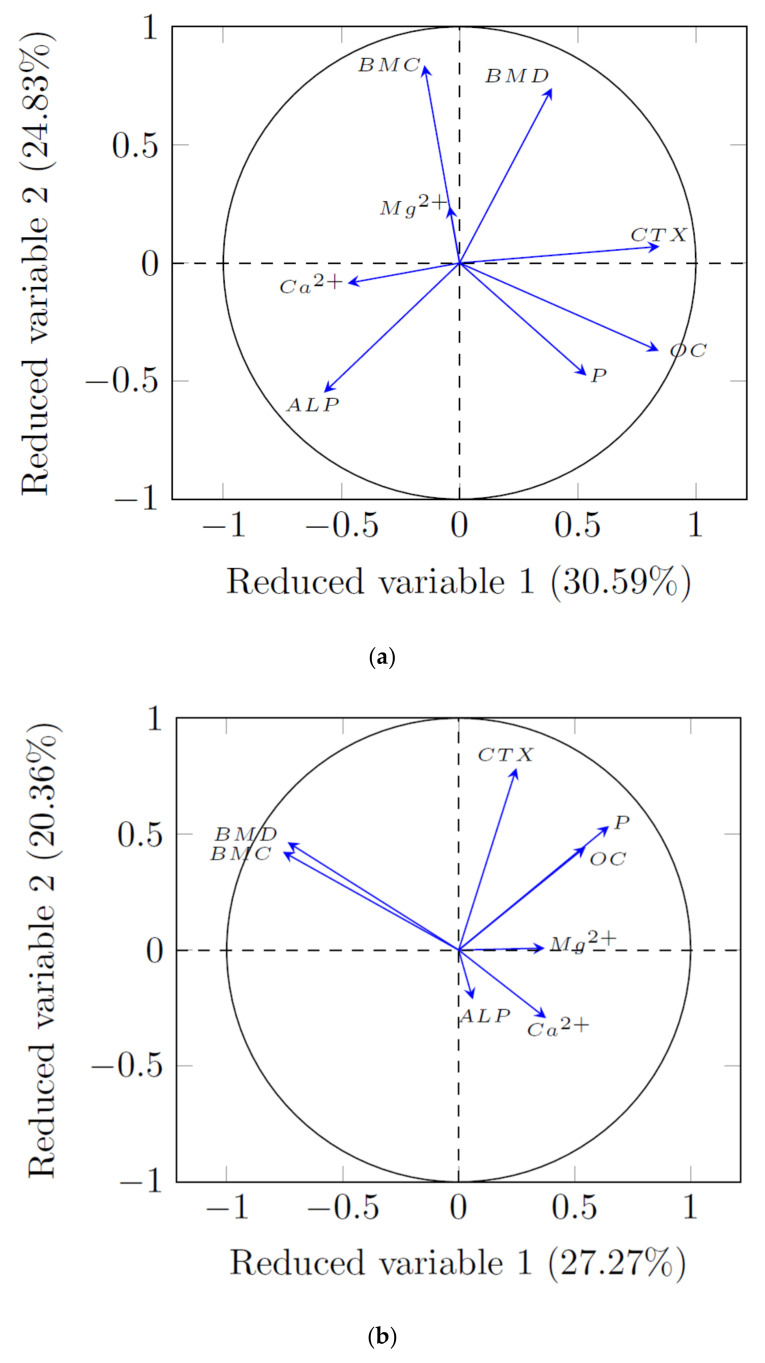
(**a**) Principal component analysis (second version, option 1) with distribution of analyzed parameters (validation model). (**b**) Principal component analysis (second version, option 2) with distribution of analyzed parameters (LC groups). (**c**) Principal component analysis (second version, option 3) with distribution of analyzed parameters (all experimental groups).

**Figure 4 animals-12-01134-f004:**
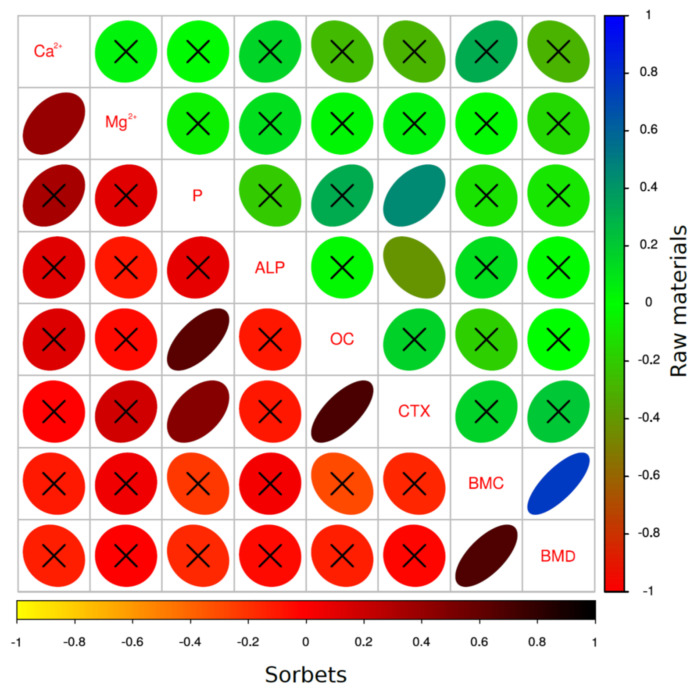
Correlations between examined parameters in terms of dependence on form of addition of fructan (raw material or sorbet) to the rats’ diet.

**Table 1 animals-12-01134-t001:** Characteristics of experimental groups.

Group Shortcut	Group Description	Number of Animals
RC	Animals fed a diet with recommended calcium dose (100% Ca)	8
LC	Animals fed a Ca-deficient diet (60% Ca)	8
LC-JA	Animals fed a Ca-deficient diet (60% Ca), with Jerusalem artichoke pulp added alone	8
LC-Y	Animals fed a Ca-deficient diet (60% Ca), with yacon root powder added alone	8
LC-F	Animals fed a Ca-deficient diet (60% Ca), with the formulation Beneo Orafti Synergy 1 added alone	8
LC-JAS	Animals fed a Ca-deficient diet (60% Ca), with Jerusalem artichoke pulp added in a strawberry sorbet matrix	8
LC-YS	Animals fed a Ca-deficient diet (60% Ca), with yacon root powder added in a strawberry sorbet matrix	8
LC-FS	Animals fed a Ca-deficient diet (60% Ca), with the formulation Beneo Orafti Synergy 1 added in a strawberry sorbet matrix	8

**Table 2 animals-12-01134-t002:** The level of selected parameters of mineral metabolism in growing rats after 12 weeks of feeding—validation model.

Parameter	Validation Group	
	RC	LC
Ca ^2+^ (mg/dL)	5.98 ^A^ ± 0.27	5.93 ^A^ ± 0.32
Mg^2+^ (mg/dL)	3.52 ^A^ ± 0.18	3.41 ^A^ ± 0.48
P (mg/dL)	5.41 ^A^ ± 1.75	7.51 ^B^ ± 2.15
ALP (mg/dL)	94.00 ^A^ ± 19.86	80.65 ^A^ ± 16.35
OC (ng/mL)	252.61 ^A^ ± 54.67	281.31 ^A^ ± 106.69
CTX (ng/mL)	14.81 ^A^ ± 2.67	24.07 ^B^ ± 11.26
BMC (g)	0.36 ^A^ ± 0.02	0.36 ^A^ ± 0.02
BMD (g/cm^2^)	0.11 ^A^ ± 0.02	0.11 ^A^ ± 0.01

^A,B^—different capital letters mean significant differences between RC and LC groups (*p* < 0.05). RC means diet with recommended calcium dose; LC means low-calcium diet (60% recommended calcium dose).

**Table 3 animals-12-01134-t003:** Effects of different fructan sources added to the diet (alone or as components of sorbet) on selected parameters of mineral metabolism in growing rats after 12 weeks of feeding.

Parameter (unit)	LC	Fructan Source					
		Type of Raw Material			Type of Sorbet		
		LC-JA	LC-Y	LC-F	LC-JAS	LC-YS	LC-FS
Ca ^2+^ (mg/dL)	5.93 ^ab^ ± 0.32	6.28 ^c^ ± 0.37	5.91 ^ab^ ± 0.35	6.06 ^bc^ ± 0.38	6.28 ^c^ ± 0.27	6.17 ^b^ ± 0.28	5.66 ^a^ ± 0.41
Mg ^2+^ (mg/dL)	3.41 ^a^ ± 0.48	3.84 ^bc^ ± 0.48	3.77 ^bc^ ± 0.17	4.12 ^c^ ± 0.54	3.59 ^ab^ ± 0.25	3.79 ^bc^ ± 0.27	3.59 ^ab^ ± 0.22
P (mg/dL)	7.51 ^c^ ± 2.15	5.60 ^a^ ± 0.79	6.01 ^ab^ ± 0.69	6.87 ^abc^ ± 0.75	5.35 ^a^ ± 1.02	6.40 ^ab^ ± 1.20	6.00 ^a^ ± 1.96
ALP (mg/dL)	80.65 ^b^ ± 16.35	99.15 ^c^ ± 11.02	96.11 ^c^ ± 16.84	73.23 ^b^ ± 15.34	70.67 ^ab^ ± 18.15	64.10 ^a^ ± 13.58	67.78 ^ab^ ± 8.89
BMC (g)	0.36 ^ab^ ± 0.02	0.37 ^ab^ ± 0.02	0.39 ^b^ ± 0.04	0.35 ^a^ ± 0.02	0.39 ^b^ ± 0.04	0.38 ^ab^ ± 0.02	0.36 ^ab^ ± 0.02
BMD (g/cm^2^)	0.11 ^a^ ± 0.01	0.12 ^a^ ± 0.01	0.12 ^a^ ± 0.03	0.11 ^a^ ± 0.02	0.12 ^a^ ± 0.04	0.12 ^a^ ± 0.02	0.11 ^a^ ± 0.02

^a,b,c^—different small letters mean significant differences among low-calcium groups (*p* < 0.05). LC: low-calcium diet, LC-JA: low-calcium diet enriched with Jerusalem artichoke, LC-Y: low-calcium diet enriched with yacon, LC-F: low-calcium diet enriched with Beneo Orafti Synergy 1, LC-JAS: low-calcium diet containing sorbet enriched with Jerusalem artichoke, LC-YS: low-calcium diet containing sorbet enriched with yacon, LC-FS: low-calcium diet containing sorbet enriched with Beneo Orafti Synergy 1.

**Table 4 animals-12-01134-t004:** The results of two-way analysis of variance (*p*-values).

Parameter	Factor 1 (Type of Fructan Source)	Factor 2 (Form of Fructan Source)	Factor 1 and Factor 2
Ca^2+^	0.006	ns	ns
Mg^2+^	0.004	0.037	ns
P	0.020	ns	ns
ALP	ns	0.000	0.005
OC	0.010	ns	ns
CTX	0.005	ns	ns
BMC	0.035	ns	ns
BMD	ns	ns	ns

ns—lack of statistical significance.

## Data Availability

The data presented in this study are available on request from the corresponding author.

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
