# Peer review of "When Incorporated into Fruit Sorbet Matrix, Are the Fructans in Natural Raw Materials More Beneficial for Bone Health than Commercial Formulation Added Alone?"

_animals, 2022, doi:10.3390/ani12091134_

Round 1
Reviewer 1 Report
The paper is very interesting and can be considered for publication in Animals. However, before publications, some topics have to be added to get the article more interesting for the readers. Indeed, no mention to oral cavity is done: fructans in very important also for animal oral hygiene. So:
- Authors should consider the importance of oral hygiene and potential mouthrinses also for animals. Please cite DOI10.23805/JO.2019.12.01.20 and PubMed ID28696070
After these changes the paper can be acceptable for publication
Author Response
Reviewer 1
- Indeed, no mention to oral cavity is done: fructans in very important also for animal oral hygiene. So: Authors should consider the importance of oral hygiene and potential mouthrinses also for animals. Please cite DOI10.23805/JO.2019.12.01.20 and PubMed ID28696070
We are grateful the Reviewer for all valuable comment, and we fully agree that oral health is a very important issue. In the literature there are a few papers concerned on the role of fructans. Example: Because of the fact that inulin is a less cariogenic substrate, inulin mouth rinsing may serve as a novel method for the control of oral malodour. (Doran, Anna & Verran, Joanna. (2007). A clinical study on the effect of the prebiotic inulin in the control of oral malodour. Microb Ecol Health Dis. 19. 10.1080/08910600701521279). However, this subject is not associated with our research area. Simultaneously, we were trying our best to cite 2 papers according to Reviewer advice:
Polizzi E., Tetè G., Bova F., Pantaleo G., Gastaldi G., Capparè P., Gherlone E. Antibacterial properties and side effects of chlorhexidine-based mouthwashes. A prospective, randomized clinical study. J Osseointegr 2020;12(1):2-7. Doi:10.23805 /JO.2019.12.01.20
Tecco S, Grusovin MG, Sciara S, Bova F, Pantaleo G, Capparé P. The association between three attitude-related indexes of oral hygiene and secondary implant failures: A retrospective longitudinal study. Int J Dent Hyg. 2018 Aug;16(3):372-379. doi: 10.1111/idh.12300. Epub 2017 Jul 11. PMID: 28696070.
Unfortunately, it was not possible, because these very interesting papers are not concerned on fructans. This is why we decided not to cite them.
Reviewer 2 Report
In the manuscript submitted to Animals entitled “The effect of fructans from different sources on bone biochemical and densitometric parameters in the model of growing female rat”, the authors presented the effect of various diets containing fructans on bone density.
I believe that this article might be interesting to the broad audience of Animals. However, major revision is needed to overcome shortcomings that are evident in experimental design and the presentation of data. Besides, significant changes are needed regarding spelling and grammar.
First of all, in the introduction, it is necessary to explain and elaborate more thoroughly on the mechanism of fructans affecting bone density. Significant changes are needed regarding the English language throughout the whole manuscript.
Regarding the materials and methods section, described experimental design is quite confusing; I would suggest a table with all experimental groups and sample sizes used. Also, in the experimental design, it is crucial to write down how old rats were when they entered the study and for how long they were fed with the specific diet.
Data presentation is quite messy and confusing; for a reader, it is easier and more comprehensible to show data as graphs than as data tables. I would suggest to show the most important data and results as graphs.
To analyze data, Principal Component Analysis was used to illustrate the correlation between experimental groups. Procedure of PCA method is quite poorly described in Statistical analysis section, so I would suggest to describe it more thoroughly (e.g. Eigen values, Scree plots,…).
Author Response
Reviewer 2
- “in the introduction, it is necessary to explain and elaborate more thoroughly on the mechanism of fructans affecting bone density”
The appropriate text was added in the Introduction section. Thank you very much for this valuable advice.
- “significant changes are needed regarding the English language throughout the whole manuscript”
The whole text was carefully checked regarding the English language. We are very sorry for all mistakes. We were trying to do our best to improve it.
- Regarding the materials and methods section, described experimental design is quite confusing; I would suggest a table with all experimental groups and sample sizes used.
We would like to thank for this valuable comment. The Table was added and the text – improved.
- Also, in the experimental design, it is crucial to write down how old rats were when they entered the study and for how long they were fed with the specific diet.
The appropriate information was added to the text.
- Data presentation is quite messy and confusing; for a reader, it is easier and more comprehensible to show data as graphs than as data tables. I would suggest to show the most important data and results as graphs.
We would like to thank the Reviewer for this comment. We decided to show 2 bone turnover parameters as graph.
- To analyze data, Principal Component Analysis was used to illustrate the correlation between experimental groups. Procedure of PCA method is quite poorly described in Statistical analysis section, so I would suggest to describe it more thoroughly (e.g. Eigen values, Scree plots,…).
We are grateful for this comment. We added appropriate text to the materials and methods section, with literature reference.
Reviewer 3 Report
To Authors
The presented manuscript contains an interesting hypothesis worth further continuation in the form of further research. Nevertheless, the publication should contain a clearly formulated hypothesis and research purpose, which was missing from this manuscript. Please fill in these mistakes. I included the rest of the minor comments in the review. In addition, I like the interdisciplinarity of the authors' team that has been used in the implementation of this and I think also in future research.

Author Response
Reviewer 3
- Nevertheless, the publication should contain a clearly formulated hypothesis and research purpose, which was missing from this manuscript.
We would like to thank the Reviewer very much for all the comments. The hypothesis and research purpose were clarified and added to the text.
- Title
I guess the title should be different. It should be changed after the correct formulation of the research hypothesis. Please suggest a different title that clearly indicates what the fructans were used for this project.
We modified the title, according to valuable advice of Reviewer.
- Line 25: I wish the bioavailability of calcium was assessed at work. Such research would greatly enrich the manuscript. Please think about it for the future.
We agree with the Reviewer that the data concerned on biovailability is of importance. For that reason, we plan a wide research concerned on this aspect in our next project. Thank you for this advice.
- Line 51-101: Throughout the entire section of Intorduction, I haven't found explanation why such biochemical and densitometric indicators were determined during the research. What is their translation into the expected research result? This is important. Please explain this in the manuscript and in the review.
The appropriate information was added in the “Introduction” section. The significance of examined parameters was explained in the section “Discussion”.
- Line 77: Please list which bioactive moieties the authors mean?
The cited paper was concerned on the bioactive compounds in a wide range. We modified the text, and added some appropriate examples.
- Line 96-101: This is not a well-articulated purpose of the research paper. I suggest establishing a research hypothesis, i.e. what the authors wanted to check, and then create the correct research goal, adequate to the laboratory analyzes carried out. In the current edition, the goal is misplaced. Please change it.
We would like to thank the Reviewer very much for this advice. We improved the manuscript, and our research hypothesis was added.
- Line 128-130: Were blood collected from all rats in the experimental groups? What volume of blood was collected from each animal? Please clarify this.
Line 132-138: Please specify in the manuscript which analytical techniques were used for biochemical analyzes.
The detailed information concerned on “Material and methods” was added/improved.
- Line 452-459: These are not the correct conclusions from the research that was carried out. The second part of the presented conclusions contains the hallmarks of conclusions, but does not answer the question why fructans protect against calcium hypoalimentation and why their action is strongly related to their source and the dietary matrix. How is it related? The conclusions should correspond to the hypothesis, which, unfortunately, is missing in the publication. Please provide specific conclusions, including practical conclusions.
According to the Reviewer’s advice, we improved the conclusions. Thank you very much for this comment and a chance to improve the manuscript.
Round 2
Reviewer 2 Report
After implementing all suggested advices and corrections, I consider this article ready for publication.
Reviewer 3 Report
The authors properly responded to my comments.